# Partial Multi-label Learning Based On Near-Far Neighborhood Label Enhancement And Nonlinear Guidance

### Yu Chen
Guangdong University of Technology
Gaungzhou, Guangdong, China
2112304132@mail2.gdut.edu.cn

### Yanan Wu
Guangdong University of Technology
Gaungzhou, Guangdong, China
2112304112@mail2.gdut.edu.cn

### Han Na
Guangdong Polytechnic Normal
University
Gaungzhou, Guangdong, China
hannagdut@126.com

### Xiaozhao Fang*
Guangdong University of Technology
Gaungzhou, Guangdong, China
xzhfang168@126.com

### Bingzhi Chen*
Beijing Institute of Technology
Zhuhai, Guangdong, China
chenbingzhi.smile@gmail.com

### Jie Wen
Harbin Institute of Technology
Shenzhen, Guangdong, China
jiewen_pr@126.com

## ABSTRACT

Partial multi-label learning (PML) deals with the problem of accurately predicting the correct multi-label class for each instance in multi-label data containing noise. Compared with traditional multi-label learning, partial multi-label learning requires learning and completing multi-label classification tasks in an imperfect environment. The existing PML methods have the following problems: (1) the correlation between samples and labels is not fully utilized; (2) the nonlinear nature of the model is not taken into account. To solve these problems, we propose a new method of PML based on label enhancement of near and far neighbor information and nonlinear guidance(PML-LENFN). Specifically, the original binary label information is reconstructed by using the information of sample near neighbors and far neighbors to eliminate the influence of noise. Then we construct a linear multi-label classifier that can explore label correlation. In order to learn the nonlinear relationship between features and labels, we use nonlinear mapping to constrain this classifier, so as to obtain the prediction results that are more consistent with the realistic label distribution.

## CCS CONCEPTS

• **Computing methodologies** → *Machine learning algorithms.*

## KEYWORDS

Partial multi-label learning, Noise elimination, Label correlations, Label enhancement, Nonlinear mapping

**ACM Reference Format:**
Yu Chen, Yanan Wu, Han Na, Xiaozhao Fang, Bingzhi Chen, and Jie Wen. 2024. Partial Multi-label Learning Based On Near-Far Neighborhood Label Enhancement And Nonlinear Guidance. In *Proceedings of the 32nd ACM*

---

*Corresponding authors: Xiaozhao Fang and Bingzhi Chen.

---

*International Conference on Multimedia (MM '24), October 28-November 1, 2024, Melbourne, VIC, Australia.* ACM, New York, NY, USA, 10 pages. https://doi.org/10.1145/3664647.3681300

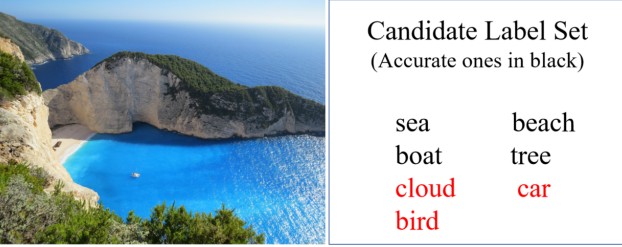

**Figure 1: Example of a picture with candidate labels. Of the candidate labels, "sea," "beach," "boat," and "tree" are real labels, while "cloud," "car," and "bird" are irrelevant labels.**

## 1 INTRODUCTION

In machine learning, objects in the real world can be abstracted into samples. A sample usually consists of two parts, the feature and the label. Labels are mutually exclusive in traditional classification tasks, which means that only one label can be tagged per sample. However, many objects in the real world have several semantic interpretations. For example, a movie may have several different themes, such as science fiction, war, and adventure; a news article can be tagged with multiple labels, such as politics, economics, and sports. Since multi-label learning (MLL)[1] can assign a set of discrete non-exclusive labels to a sample, it has aroused great research interest in the field of machine learning, and is widely used in various fields of real scenes, such as protein classification[2], image annotation[3, 4], gene function prediction[5], etc. However, in practical scenarios, acquiring datasets with accurate annotation labels is a costly and formidable task. Typically, one can only access a set of candidate labels, comprising relevant ones as well as noise labels. For instance, in Figure 1, due to the imprecise handling by the noisy annotator, the picture is associated with "sea," "beach," "boat," and "tree" among the candidate labels; while the remaining labels denote noise. Therefore, the presence of unrelated labels in the candidate labels may adversely affect the performance of multi-label learning. Nonetheless, the effects of noise can be mitigated by removing noise labels such as "cloud", "car", and "bird" from candidate labels during training, or by ignoring certain obscure features.

In addition, label correlation helps identify underlying truth labels. For example, in Figure 1, the label "sea" often appears with the label "boat". To address this challenge, Xie and Huang[6] firstly proposed partial multi-label learning (PML) as a novel framework. PML represents a form of weakly supervised learning, where the essential fact label for each instance remains concealed within the candidate labels and cannot be directly accessed. Its objective is to train a model capable of predicting labels for unknown instances using a multi-label dataset containing noisy information. This innovative approach, not only provides an effective solution but also garners significant attention in the field.

The main challenge facing PML is how to effectively identify noise labels and build a classification model with higher accuracy, while some traditional multi-label learning algorithms do not process noise labels in candidate labels. Examples include RankSVM[5], ML-kNN[7] and LIFT[8], which handle label noise poorly and are inadequate for PML work. Later developed PML methods generally categorize into two strategies: the two-stage strategy and the integrated end-to-end approaches[9].

In the two-stage strategy, the process is bifurcated: first, authentic labels are sifted from the candidates, followed by classifier training with these vetted labels using sophisticated Multi-Label Learning (MLL) techniques. PARTICLE[10] leveraged label propagation to surface confident labels. PAMB[11] utilized ECOC to reformulate PML as binary problems, bypassing direct label confidence estimation. DRAMA[12] assessed label reliability based on feature manifolds and trained with gradient-boosted decision trees, informed by this reliability. Xu et al[13, 14] proposed enhancing labels by recovering per-instance distributions, using label correlations and feature space topology.

The end-to-end strategy concurrently optimizes candidate labels and model training. PML-fp and PML-lc[6] constrained label relationships to estimate confidence for predictive modeling. PML-LRS[15] applied low-rank and sparse decomposition to isolate true labels from noise. MUSER[16] tackled feature noise, learning from feature and label subspaces to reduce bias. PML-MD[17] employed a meta-learning approach to disambiguation within a distinct setting.

The above methods are all proposed for PML problems. However, while these methods have solved some difficult problems from many different aspects, there are still some problems that need to be further explored. (1) The correlation between samples and labels is not fully utilized. Since the candidate labels in PML are not trusted, the disambiguation based on matrix decomposition and the recognition based on feature information can eliminate the natural noise, but the correlation of labels is ignored, and the effect of dealing with artificial noise is not good. In the method based on label confidence, nearest neighbor information or clustering information is usually used to obtain the label confidence and remove the noise label. However, for samples lacking or away from neighbors, it is often difficult to choose the right confidence threshold to achieve better results. And the relationship between samples and labels is not fully utilized. (2) The nonlinear characteristics of the model are not considered. MLL's task is to train a multi-label classifier that can assign an appropriate set of labels to unseen instances. At present, most of the multi-label predictors have linear structure, but in practice, the models we need to fit are basically curvilinear structure, that is, nonlinear structure, and it is difficult to fit the real

nonlinear model with only the linear structure model. Considering the above points, we propose a new method of PML based on label enhancement of near and far neighbor information and nonlinear guidance(PML-LENFN). In the first part, in order to deal with the situation that the sample lacks a neighbor or is still far away from its nearest neighbor, we add the far neighbor information as the reference information, and reconstruct the original label according to its own information, near neighbors information and far neighbors information. The refactored label is no longer a binary label, and the model is then trained using the reconstructed label set. In the second part, we construct a linear multi-label classifier. In order to explore the relationship between labels, we guide the classifier learning through the label Laplacian matrix. In the third part, the prediction results of linear classifier are decomposed, and nonlinear mapping is added to learn the nonlinear relationship between features and labels, so that the prediction results are more consistent with the real label distribution. The main contributions are summarized as follows:

- We propose a new label enhancement method, which reconstructs label information by using both near and far information of samples to achieve denoising effect.
- The combination of linear and nonlinear training method is used to train the classifier, which not only retains the linear characteristics of the model, but also adds the nonlinear characteristics to make it more consistent with the actual results.
- Extensive experiments and analyses conducted on 3 real PML datasets and 17 synthetic PML datasets demonstrate the superiority of the proposed PML-LENFN method over existing methods.

The remainder of this article is structured as follows: In section 2, we provide an overview of the research on multi-label learning, partial label learning, and partial multi-label learning. Section 3 delves into a detailed explanation of the PML-LENFN principle and its potential optimizations. Section 4 showcases the results from a series of experiments and conducts an analysis. Finally, in Section 5, we summarize this thesis.

## 2 RELATED WORK

In this section, we give a brief overview of the work related to partial multi-label learning, as well as the closely related multi-label learning and partial label learning.

### 2.1 Multi-label learning

In the realm of Multi-Label Learning (MLL), each instance is connected to multiple precise labels, a subject extensively explored in research. Traditional methods often converted MLL into binary classification tasks[18, 19], treating each label in separation. Yet, to improve outcomes, studies have increasingly focused on utilizing the interplay among labels, ranging from pairwise correlations[20] to comprehensive higher-order label relationships.

In recent years, the integration of manifold learning and multi-label learning has garnered significant attention. Leveraging the assumption that samples with strong correlations may share labels, Geng et al[21] investigated the manifold structure preserved in label space. Meanwhile, Luo et al[22] utilized low-dimensional

embedding to construct label information based on manifold learning and sparse feature selection. To mitigate the impact of noise and missing labels, Zhao et al[23] combined manifold learning and subspace learning to reconstruct potential feature space and label space unaffected by feature noise and missing labels. Furthermore, in recent years, numerous scholars [24, 25] have incorporated multiview [26] into multi-label learning to capture more comprehensive sample information. It is important to note that MLL assumes accurate labeling for each instance, which is nearly unattainable in real-world scenarios. Obtaining high-precision labeled datasets is indeed challenging.

## 2.2 Partial label learning

Partial Label Learning (PLL) is distinguished by instances each paired with a set of potential labels, of which only one is the true label[27–29]. The primary approach to PLL involves disambiguation, seeking to uncover the singular ground-truth label from the set. Certain methodologies predict outcomes by averaging the potential labels' outputs[30, 31], where each instance is given uniform consideration. In contrast, other strategies incorporate the ground-truth label as a latent element, employing an iterative process to enhance model parameters[28, 32–34]. PLL and PML both operate on training data that lacks precise labeling, with instances each receiving a set of candidate labels. While some PLL algorithms excel in handling noisy datasets, they are confined to single-label predictions, which poses challenges when applied to multi-label datasets[35].

## 2.3 Partial multi-label learning

As mentioned above, Compared with MLL and PLL, partial multi-label learning(PML) is more challenging, requiring both learning in imperfect environments and training high-precision classifiers.

Recently, numerous algorithms have been developed to tackle the issue of partial multi-label learning. PML-lc and PML-fp [6] utilized confidence values for acquiring ground-truth labels and training the classifier. The methodologies of PML-VLS and PML-MAP [36] extracted reliable labels and conducted model induction from the candidate label set through an iterative label propagation process. In a study by Sun et al[15], a low-rank and sparse decomposition scheme was employed to learn the prediction model. Several other approaches aim to address the PML problem by incorporating feature information. For instance, fPML [37] leveraged dependencies between labels and features to distinguish ground-truth labels from outliers while training the classifier. In another recent work DRAMA [12], emphasis was placed on utilizing feature manifold to solve the PML problem. Additionally, in PML-NI [38], it was assumed that noisy labels are often caused by ambiguous contents in examples, addressing the PML problem through decomposing the prediction model matrix into ground-truth label predictions and identification of noisy labels. Furthermore, in PAMB [11], the task of learning in partial multi-label setting was transformed into multiple binary learning problems using error-correcting output codes (ECOC) techniques, avoiding estimation of labeling confidence for individual candidate labels which is prone to errors. Lastly, MUSER [16] trained a robust PML model considering noise in both feature space and label space. The study by HALE

[39] clarified the set of candidate labels and pinpointed reliable labels for training instances by leveraging correlations between instance-label assignments. PML-SALC [40], a novel approach in partial multi-label learning, proposed that label correlations should be both asymmetric and sparse, utilizing global asymmetric cues and feature structural patterns to discern these relationships. Such correlations were instrumental in mitigating the impact of noisy labels. PML-DNDC [41] introduced a groundbreaking PML strategy, pioneering a dual noise cancellation technique that tackled both label and feature noise simultaneously. This method also bolstered classifier training by dynamically uncovering latent label interdependencies, encouraging a scenario where related labels inclined towards converging on similar classifiers.

Different from the above methods, our method not only considers the relationship between labels and samples by using the information about the distance of samples, but also explores the correlation between labels to help the classifier training. We also take into account the nonlinear nature of the model to aid in the final prediction results. The main framework of PML-LENFN is shown in Figure 2.

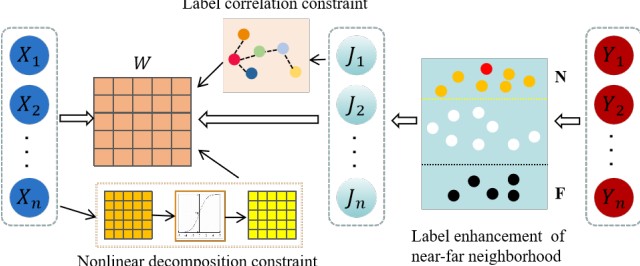

**Figure 2: The main framework of PML-LENFN, where N and F represent the domain of near and far neighborhood**

## 3 PROPOSED METHOD

In this section, we delve into the specifics of PML-LENFN. Define symbols in some articles: $X = [x_1, x_2, \ldots, x_n] \in \mathbb{R}^{d \times n}$ signifies the feature matrix encompassing $n$ instances each with $d$-dimensional features. $Y \in \{0, 1\}^{n \times c}$ indicates the candidate label matrix encompassing $n$ instances each with tagged the $c$ class labels. If $Y_{ij} = 1$, it means that the $i$-th instance is annotated with the $j$-th label. If $Y_{ij} = 0$, the converse implies no such association. The objective of PML is to mitigate noise impact and ensure accurate label predictions.

## 3.1 Label enhancement based on near and far neighbors

Since PML labels lack trustworthiness, several existing methods attempt to transform the candidate label set into trusted labels by leveraging neighbor-based approaches. These methods operate under the assumption that as the similarity between samples increases, their corresponding labels should also show a greater degree of similarity. However, in datasets with sparse neighbors, the efficacy of such methods diminishes, with instances where even the nearest neighbors are distantly located from the focal samples. Acknowledging this challenge, we incorporate information from far

neighbors, positing that as samples become less similar, their labels should also demonstrate decreased similarity. In order to reduce the situation that the nearest neighbor is still very far, we avoid the traditional KNN method, and choose to calculate the sample similarity and establish a threshold to select the near neighbor and far neighbor. The formula for calculating similarity score using the heat kernel function is as follows:

$$S_{ij} = exp(\frac{-||x_i - x_j||_2^2}{2\sigma^2}) \tag{1}$$

Where the parameter $\sigma = \sum_{i=1}^{n} ||x_i - x_{ik}||_2/n$, and $X_{ik}$ is the k-th nearest neighbor of $X_i$. The thermonuclear function returns a similarity score between 0 and 1, with close to 1 indicating that the samples are very close, and close to 0 indicating that the samples are far apart [42, 43]. Then we set thresholds to pick the near and far neighbors of the samples, if the similarity between the j-th sample and the i-th sample is greater than a certain threshold, it is listed as the near neighbors of the i-th sample, and if it is less than a certain threshold, it is listed as the far neighbors of the i-th sample. The corresponding formula is as follows:

$$\begin{cases} N_i = N_i \cup \{x_j\}, & if\ S_{ij} > \tau_n^{(i)} \\ F_i = F_i \cup \{x_j\}, & if\ S_{ij} < \tau_f^{(i)} \\ N_i = N_i, F_i = F_i, & otherwise. \end{cases} \tag{2}$$

Where $N_i$ represents the near neighbors of the i-th sample and $F_i$ represents its far neighbors, $\tau_n$ and $\tau_f$ represent similarity thresholds of near and far neighbors. Because the similarity distribution of samples in different data sets is different, the threshold values of each data set are different or even very different. In order to avoid the difficulty in finding the optimal absolute threshold, we choose the relative threshold to select the nearest and far neighbors. This method is expressed as the following formula:

$$\begin{cases} \xi_i = \mu \cdot (\max(S_{i\cdot}) - \min(S_{i\cdot})) \\ \tau_n^{(i)} = \max(S_{i\cdot}) - \xi_i \\ \tau_f^{(i)} = \min(S_{i\cdot}) + \xi_i \end{cases} \tag{3}$$

Where $\mu \in [0, 1]$ is the relative distance regulatory factor. In order not to over-reference the information of the near and far neighbors, we reconstruct the new label information by combining the label information of the samples themselves and the label information of their near and far neighbors. The corresponding formula is as follows:

$$J_{i\cdot} = \alpha Y_i + (1 - \alpha - \beta)\bar{Y}_n + \beta(1 - \bar{Y}_f)$$
$$s.t. \quad x_n \in N_i, \quad x_f \in F_i \tag{4}$$

Where $J_{i\cdot}$ represents the label information of i-th sample after reconstruction, the reconstructed label $J \in [0, 1]^{n \times c}$, $\bar{Y}_n$ and $\bar{Y}_f$ represent the sum and average of the label information corresponding to the near and far neighbors.

## 3.2 Linear classifier based on label correlation

For forecasting pertinent labels of new instances, the pivotal task is to cultivate the classifier matrix $W = [w_1, w_2, \ldots, w_c]^T \in \mathbb{R}^{d \times c}$ facilitating the mapping from instances to their corresponding labels. In this context, the classifier is trained directly on the refined labels to mitigate the influence of noise, bypassing the use of preliminary candidate labels. Consequently, the linear classifier $W$ is

trained using matrix $J$. The formulation is crafted by drawing upon conventional machine learning principles, as delineated below:

$$\min_W ||J - X^TW||_F^2 + \Phi(W) \tag{5}$$

Where $\Phi(W)$ is defined as a regularization function of $W$, which controls the complexity. Simultaneously, one cannot overlook label correlation in multi-label learning, as it can significantly enhance model performance. To mitigate noise interference, the reconstructed matrix $J$ is selected for calculating label correlation. Theoretically, label correlation is indicated by the frequency of label co-occurrence. In other words, labels exhibit stronger correlation when they appear together more frequently[44]. The reconstructed label information is not binary label, it contains more label information and better results can be obtained by using it to calculate the correlation between the labels. However, label correlation manifests asymmetrically in reality. The correlation between labels a and b may not mirror that between labels b and a. The label correlation matrix $A$ is computed as follows:

$$A_{ij} = \frac{J_{\cdot i}^T J_{\cdot j}}{||J_{\cdot i}||_2^2} \tag{6}$$

where $A$ means correlation between labels $i$ and $j$, and $J_{\cdot i}$ represents the $i$-th column of $J$. The next step involves transferring this label correlation to the classifier matrix $W$ to enhance model prediction accuracy. In essence, higher correlation between labels indicates greater similarity among their respective classifiers, and conversely. Thus, we incorporate the concept of Laplacian eigenmaps (LE)[45], which facilitates the transfer of label correlation information to the classifier matrix. This concept is expressed as follows:

$$\frac{1}{2}\sum_{i=1}^{l}\sum_{j}^{l} A_{ij}||w_i - w_j||_2^2 = Tr(WLW^T) \tag{7}$$

Where $Tr(\cdot)$ is the trace of the matrix, and $L = D - A$ is defined as graph Laplacian matrix. $D$ represents a diagonal matrix, of which $D_{ii} = \sum_{j=1}^{n} A_{ij}$. By this method, the classifier matrix $W$ is adept at performing classification tasks more effectively, leveraging asymmetric label relationships while mitigating noise interference. Recognizing the interplay of label correlations, the classifier matrix $W$ is designed to be linearly correlated to precisely reflect these interdependencies, leading to a low-rank matrix structure for $W$. Nonetheless, optimizing such a structure is complex due to the discrete nature of the rank function. Studies and empirical data indicate that the Frobenius norm is an effective convex proxy for imposing low-rank constraints[46]. The modified formula is detailed as follows:

$$\min_W ||J - X^TW||_F^2 + \lambda_1||W||_F^2 + \lambda_2 Tr(WLW^T) \tag{8}$$

## 3.3 Matrix nonlinear decomposition

Let us consider the hypothetical existence of a perfect label matrix $\tilde{Y}$, encompassing all labels with precision and completeness. This matrix can be factorized into the multiplication of two matrices, both characterized by low rank, as articulated by the following expression:

$$\tilde{Y}_{n \times l} = G_{n \times k} U_{k \times l} (k \leq c) \tag{9}$$

In this scenario, $G$ denotes a latent label matrix, which can be interpreted as a low-rank representation derived from reducing the label dimensions within $\tilde{Y}$. It is evident that $G$ is more succinct and generalized compared to $\tilde{Y}$ and it is capable of filtering out the noise present in $\tilde{Y}$. The matrix $U$ delineates the linkage between the pristine label matrix and the latent one, earning the title of the correlation matrix for $G$ [47]. Define the following objective function as:

$$\min_{G,U} ||\tilde{Y} - GU||_F^2 \qquad (10)$$

By minimizing the reconstruction error, a better latent label matrix $G$ and its association matrix $U$ can be found. The latent label matrix $G$ is low-dimensional, which is equivalent to the reduced dimension of the label matrix $\tilde{Y}$. In most multi-label learning, it is usually assumed that there is a linear mapping relationship between features and the label matrix $G$, namely:

$$G = X^T V \qquad (11)$$

Where $V \in \mathbb{R}^{d \times k}$. Yet, linear models frequently fail to capture the complexities of real-world data [48]. From equation (10), we discern that the label space $G$ is situated within the continuum of real numbers, typically ranging from $[0, 1]$. The Sigmoid function, with its smooth, S-shaped curve, maps any input to a value within the same interval, making it well-suited for multi-label learning due to its continuous and differentiable properties. It enhances the discriminative power of label predictions and facilitates a nonlinear transformation that leverages each sample's information [49]. The Sigmoid function is defined as:

$$\varphi(K) = \frac{I}{I + e^{-K}} \qquad (12)$$

Where $I \in \mathbb{R}^{n \times k}$ represents the matrix with all elements of 1, The decomposition of the label matrix is not unique. The Sigmoid function can project the result into the interval $[0, 1]$, aligning closely with the probabilistic nature of label distributions and enhancing the characterization of the interplay between features and labels. By leveraging matrix transformations, we can devise a nonlinear model that encapsulates the intricate associations between features and labels. The mathematical expression for this model is presented hereinafter:

$$G = \varphi(X^T V) = \frac{I}{I + e^{-X^T V}} \qquad (13)$$

Combined with Formula (9), the ideal label matrix $\tilde{Y}$ can be expressed as:

$$\tilde{Y} = GU = \varphi(X^T V)U = \frac{I}{I + e^{-X^T V}}U \qquad (14)$$

where $U$ and $V$ can be obtained by solving the following optimization problem:

$$\min_{U,V} ||\tilde{Y} - \varphi(X^T V)U||_F^2 + \lambda_3||U||_F^2 + \lambda_4||V||_F^2 \qquad (15)$$

First, the ideal label matrix is decomposed into the latent label matrix and association matrix. Then, in order to describe the relationship between nonlinear features and labels, the Sigmoid function is used for nonlinear mapping. By processing samples point by point, the sample information can be more effectively utilized, and the predicted output is more consistent with the real label distribution.

## 3.4 Overall Objective Function

Combining formula (8) and formula (15), in order to learn the nonlinear relationship of the model, the result of the linear classifier is taken as the ideal label in the nonlinear classifier, the final objective function of PML-LENFN is as follows:

$$\min_{W,U,V} ||J - X^T W||_F^2 + ||X^T W - \varphi(X^T V)U||_F^2 + \lambda_1||W||_F^2 + \lambda_2 Tr(WLW^T) + \lambda_3||U||_F^2 + \lambda_4||V||_F^2 \qquad (16)$$

## 3.5 Optimization

For the above optimization problem, ADMM [50] method is adopted to solve it. In other words, a variable is solved and updated by fixing the other variables. Due to the complexity of two-level nonlinear mapping, variables $H$ and $Q$ are introduced to simply express the gradient of each variable in the objective function.

$$H = \varphi(X^T V) \qquad (17)$$

$$Q = \frac{e^{-X^T U}}{(I + e^{-X^T U})^2} \qquad (18)$$

Where $I \in \mathbb{R}^{n \times k}$ represent the matrix with all elements of 1. (1) Update $W$, Fixed $U$ and $V$, after derivation and making the derivative result to be zero, we simplify it as follows:

$$2XX^T W + W(\lambda_1 + \frac{1}{2}\lambda_2 L^T + \frac{1}{2}\lambda_2 L) = XHU + XJ \qquad (19)$$

We discover that formula (18) satisfies the Sylvester equation form of $MW + WN = T$, where $M = 2XX^T, N = \lambda_1 I + \lambda_2 L^T + \lambda_2 L$ and $T = XHU + XJ$. It can be solved by lyap function in MATLAB.

(2) Update $U$, Fixed $W$ and $V$, after derivation and making the derivative result to be zero, we simplify it as follows:

$$U = (\lambda_3 + H^T H)^{-1} H^T * (X^T W) \qquad (20)$$

(3) Update $V$, Fixed $W$ and $U$, Formula (16) becomes as:

$$\min_V \Gamma(V) = \min_V ||X^T W - \varphi(X^T V)U||_F^2 + \lambda_4||V||_F^2 \qquad (21)$$

Then, the gradient of Formula (21) for can be expressed as:

$$\frac{\partial \Gamma(V)}{\partial V} = X(Q \odot ((HU - X^T W)U^T) + \lambda_4 V \qquad (22)$$

## 4 EXPERIMENT

### 4.1 Dataset

To evaluate the generalization performance of our proposed PML-LENFN method, a total of 9 datasets were used for comparative study. Specifically, the experiments were conducted on 3 real-world PML datasets and 17 synthetic PML datasets generated from 6 multi-label datasets. Detailed characteristics of all datasets are summarized in Table 1. For 3 real-world PML datasets including Music_emotion[51], Music_style[51] and YeastBP [51], whose candidate labels are concentrated with natural noise. For public MLL datasets including Genbase[52], CAL500 [53], Bibtex[54], Medical[55], Birds[56] and Emotions[57]. We generate multiple synthetic PML data sets from each of them by randomly selecting irrelevant labels to form candidate label sets alongside their ground truth labels. As illustrated in Table 1, each value in the 'num − r' column corresponds to a distinct configuration involving the selection of varying numbers of irrelevant labels, $r \in \{1, 2, 3\}$ means

**Table 1: Basic information about three real-word partial multi-label data sets and six multi-label data sets.**

| Datsets | $|Instance|$ | $|Dim|$ | Class | $num-r$ | $avg.\#GLS$ | Field |
|---------|--------------|---------|-------|---------|-------------|-------|
| Music_emotion | 6833 | 98 | 11 | | 2.42 | Music |
| Music_style | 6839 | 98 | 10 | | 1.44 | Music |
| YeastBP | 6139 | 6139 | 217 | | 5.537 | Biology |
| Genbase | 662 | 1186 | 27 | 1,2,3 | 1.252 | Biology |
| CAL500 | 502 | 68 | 174 | 1,2,3 | 26.044 | Music |
| Bibtex | 7395 | 1836 | 159 | 1,2,3 | 2.402 | Text |
| Medical | 978 | 1449 | 45 | 1,2,3 | 1.245 | Text |
| Birds | 645 | 260 | 19 | 1,2,3 | 1.014 | Audio |
| Emotions | 593 | 72 | 6 | 1,2 | 1.869 | Music |

that 1, 2, 3 random noises are injected respectively, Since Emotions data set has only six labels, and most of the true number of labels is three, So only set $r \in \{1, 2\}$, finally resulting in a total of 17 synthetic PML data sets being generated.

## 4.2 Experimental Setting

Due to space constraints, we only present the experimental results of three evaluation indicators commonly used in multi-label classification models: *Ranking Loss*, *One Error*, and *Average Precision*. For the first two evaluation indicators, the smaller the value, the better the effect, the latter is the opposite. For a detailed description of the metrics, see [58].

In order to reduce randomness, we used 10 cross-checks, and the experimental results included the mean and standard deviation of each result. We use the following eight state-of-the-art methods as benchmarks. These include two MLL algorithms, ML-KNN [7] and LIFT [59], six PML algorithms included PML-DNDC [41]. PML-SALC [40], PAR-MAP and PAR-VLS [36], fPML [37], and PML-fp [6]. Each configured with parameters suggested in respective literature.

## 4.3 Experimental Result

The results are presented in Tables 2, 3, 4 and 5. The best results of the experiment are shown in bold. Some results are from the reference[41]. It should be noted that PML-SALC was not included in the experiments for the Music emotion, Music style, and YeastBP datasets due to the absence of publicly available code. Upon reviewing the outcomes, the following insights can be deduced:

- It turns out from Table 2 that we conduct 9 cases (3 datasets × 3 metrics = 9 cases) of experiments on real-world PML datasets and PML-LENFN performs best in all cases, accounting for 100.00%. This proves that PML-LENFN performs best on these real-world datasets.
- It turns out from Tables 3, 4 and 5 that we conduct 51 cases (17 datasets × 3 metrics = 51 cases) of experiments on synthetic PML datasets where $r \in \{1, 2, 3\}$ and PML-LENFN performs best in 46 cases, accounting for 90.20%. It proves that PML-LENFN still performs best in these synthetic datasets.
- MLL algorithm works well in data sets with relatively little noise, PML algorithm is much better than MLL algorithm when there is relatively much noise. These PML algorithms are generally superior to MLL algorithms.

Following the initial analysis, we employed the post hoc Nemenyi test (at 0.05 significance level) to scrutinize the variances among the algorithms. In this study, PML-LENFN serves as the benchmark. The critical difference (CD) was calculated to be 2.9138. The assessment of statistical significance hinges on whether the gap between PML-LENFN's average ranking and that of other algorithms exceeds the CD threshold. A visual representation of these comparisons is provided in Figure 3. Upon examining the rankings across various metrics, PML-LENFN emerged as the top performer, significantly outclassing PML-fp, PAR-MAP, PAR-VLS, ML-KNN and LIFT across all evaluation criteria.

## 4.4 Parameter Analysis

The sensitivity analysis of parameters elucidates their individual influence on the model's performance. Conducted on the Emotions dataset with $r = 1$. We set the range of the four parameters $\lambda_1$, $\lambda_2$, $\lambda_3$ and $\lambda_4$ as: $\{0.001, 0.01, 0.1, 1, 10, 100\}$. In addition, there are some sensitivity parameters in this paper that need to be adjusted, the value range of parameter $k$ is :$\{1, 2, 3, 4, 5\}$, the value range of parameter $\alpha$ is:$\{0.5, 0.6, 0.7, 0.8, 0.9\}$, the value range of parameter $\beta$ is:$\{0.1, 0.2, 0.3, 0.4, 0.5\}$. Other parameters are unchanged when one parameter changes. The experimental results are shown in the Figure 4. Upon analyzing these result, we discovered that, each constraint in the model has improved effect compared with no addition. Some analysis is as follows: $\lambda_1$ represents the weight of the overfit of the equilibrium model, and its value cannot be too large or too small. $\lambda_2$ represents the label correlation constraint weight, excessive value of $\lambda_2$ will skew model focus. Adding strong constraints to labels that have little relationship with each other will reduce the performance of the model. $\lambda_3$ represents the constraint weight of the dimensionality reduction mapping matrix, and $k$ represents the dimensionality after dimensionality reduction. In practical experiments, as long as the dimensionality after dimensionality reduction is not particularly small, the model can maintain good performance, so it is not sensitive to parameters $\lambda_3$ and $k$. $\lambda_4$ gauges nonlinearity, too high can overemphasize nonlinear traits. $\beta$ represents the far neighbors' influence, too high a value may over-rely on far data points.

## 4.5 Ablation Study

In order to prove the validity of each constraint in the model, we will compare the algorithm with the degenerate algorithms. We performed ablation experiments on synthetic PML datasets with $r = 1$ random noise added to the Emotions and Birds data sets. To verify the effect of label enhancement on the model, we use the original label information and leave the rest unchanged, called LE-free. Due to space limitations, and relevant tags and neighbor algorithm has been a lot of algorithm is proved to be effective [7, 9, 36, 44], so here only proves that more far neighbors to the model. $\beta$ is set to 0 means far neighbors is not used, and the rest is unchanged, which is called Far Neighbor-free. In order to prove the effect of Nonlinear matrix decomposition on the model, we remove the constraint of the second matrix in formula (15), called Nonlinear-free. The results of ablation experiments are shown in the Figure 5. The results show that PML-LENFN is superior to all degradation algorithms, proving that every constraint in the model has a positive promotion effect on the model, among which label enhancement has the best promotion effect. And nonlinear constraints also have a promotion effect, proving that the model has partial nonlinear properties.

**Table 2: Comparision of PML-LENFN with other state-of-the-art PML and MLL algorithms on real-world datasets (mean±std),where the best result are in bold and the suboptimal is shown with an underscore.**

| Datastets | PML-LENFN | PML-DNDC | PAR-MAP | PAR-VLS | fPML | PML-fp | LIFT | ML-KNN |
|---|---|---|---|---|---|---|---|---|
| *Ranking Loss* (the smaller, the better) | | | | | | | | |
| Music_emotion | **.230±.011** | .233±.004 | .245±.006 | .261±.007 | .331±.008 | .276±.007 | .276±.009 | .302±.009 |
| Music_style | **.135±.006** | .139±.006 | .161±.006 | .161±.005 | .224±.007 | .146±.005 | .202±.010 | .199±.008 |
| YeastBP | **.178±.004** | .192±.004 | .283±.040 | .935±.024 | .415±.057 | .363±.041 | .316±.054 | .408±.060 |
| *One Error* (the smaller, the better) | | | | | | | | |
| Music_emotion | **.441±.013** | .497±.011 | .474±.018 | .473±.019 | .592±.012 | .540±.018 | .554±0.022 | .544±.018 |
| Music_style | **.350±.017** | .351±.010 | .450±.034 | .370±.016 | .404±.012 | .406±.017 | .407±.017 | .384±.015 |
| YeastBP | **.408±.006** | .435±.013 | .912±.054 | .906±.054 | .980±.015 | .992±.036 | .913±.019 | .953±.048 |
| *Average Precision* (the larger, the better) | | | | | | | | |
| Music_emotion | **.625±.011** | .608±.005 | .614±.007 | .605±.006 | .520±.008 | .567±.010 | .569±.008 | .555±.007 |
| Music_style | **.738±.010** | .734±.006 | .716±.010 | .716±.010 | .654±.007 | .703±.009 | .665±.008 | .683±.009 |
| YeastBP | **.460±.018** | .376±.007 | .158±.019 | .086±.019 | .095±.020 | .143±.021 | .169±.058 | .110±.029 |

**Table 3: Comparision of PML-LENFN with other state-of-the-art PML and MLL algorithms on synthetic datasets and three evaluation metrics when $r = 1$ (mean±std), where the best results are in bold and the suboptimal are shown with an underscore.**

| Datasets | PML-LENFN | PML-DNDC | PML-SALC | PAR-MAP | PAR-VLS | fPML | PML-fp | LIFT | ML-KNN |
|---|---|---|---|---|---|---|---|---|---|
| *RankingLoss* (the smaller, the better) | | | | | | | | | |
| Genbase | **.001±.002** | .003±0.02 | .005±.003 | .014±.006 | .025±.010 | .0170.006 | .031±.013 | .009±.007 | .008±.006 |
| CAL500 | **.173±.010** | .181±.004 | .178±.008 | .176±.005 | .538±.010 | .181±.005 | .182±.009 | .184±.007 | .185±.007 |
| Bibtex | **.073±.004** | .075±.003 | .107±.008 | .320±.007 | .325±.010 | .105±.009 | .108±.007 | .110±.005 | .225±.005 |
| Medical | **.021±.007** | .027±.014 | .025±.006 | .080±.013 | .106±.018 | .050±.012 | .098±.019 | .036±.008 | .061±.012 |
| Birds | .172±.032 | .173±.017 | **.170±.032** | .316±.022 | .319±.019 | .226±.023 | .426±.040 | .263±.029 | .336±.025 |
| Emotions | **.163±.036** | .183±.021 | .193±.020 | .230±.051 | .242±.016 | .206±.014 | .386±.021 | .263±.027 | .312±.026 |
| *One Error* (the smaller, the better) | | | | | | | | | |
| Genbase | **.000±.000** | .000±.002 | .019±.009 | .028±.015 | .050±.024 | .030±.014 | .043±.010 | .027±.013 | .024±.014 |
| CAL500 | .117±.014 | .126±.027 | .119±.022 | **.116±.019** | .324±.051 | 0.120±.019 | .130±.008 | .124±.024 | .116±.019 |
| Bibtex | **.360±.009** | .363±.009 | .371±.011 | .743±.017 | .588±.012 | .453±.017 | .377±.006 | .401±.017 | .624±.012 |
| Medical | **.137±.037** | .153±.028 | .164±.041 | .441±.073 | .252±.051 | .194±.038 | .345±.043 | .174±.039 | .276±.035 |
| Birds | **.390±.076** | .399±.059 | .401±.046 | .689±.036 | .618±.047 | .669±.033 | .859±.058 | .777±.081 | .738±.036 |
| Emotions | **.276±.067** | .309±.048 | .326±.052 | .355±.064 | .288±.048 | .349±.040 | .469±.042 | .386±.048 | .419±.047 |
| *Average Precision* (the larger, the better) | | | | | | | | | |
| Genbase | **.995±.007** | .993±.005 | .991±.004 | .980±.011 | .959±.022 | .981±.010 | .985±.008 | .982±.011 | .983±.010 |
| CAL500 | **.516±.013** | .514±.007 | .513±.014 | .511±.011 | .380±.019 | .504±.015 | .506±.011 | .504±.015 | .490±.011 |
| Bibtex | **.595±.001** | .589±.007 | .572±.003 | .477±.008 | .482±.007 | .546±.008 | .546±.008 | .528±.014 | .321±.006 |
| Medical | **.888±.032** | .885±.020 | .871±.033 | .651±.051 | .733±.034 | .833±.029 | .694±.041 | .855±.029 | .772±.025 |
| Birds | **.632±.050** | .629±.036 | .629±.041 | .402±.025 | .408±.025 | .433±.029 | .273±.027 | .360±.044 | .371±.021 |
| Emotions | **.798±.043** | .777±.024 | .769±.029 | .736±.030 | .758±.017 | .752±.019 | .626±.020 | .703±.025 | .672±.023 |

**Table 4: Comparision of PML-LENFN with other state-of-the-art PML and MLL algorithms on synthetic datasets and three evaluation metrics when $r = 2$ (mean±std), where the best results are in bold and the suboptimal are shown with an underscore.**

| Datasets | PML-LENFN | PML-DNCN | PML-SALC | PAR-MAP | PAR-VLS | fPML | PML-fp | LIFT | ML-KNN |
|---|---|---|---|---|---|---|---|---|---|
| *Ranking Loss* (the smaller, the better) | | | | | | | | | |
| Genbase | **.002±.002** | .003±.002 | .005±.002 | .018±.007 | .024±.007 | .010±.005 | .008±.003 | .010±.005 | .013±.009 |
| CAL500 | .177±.009 | .180±.007 | .178±.014 | **.170±.005** | .553±.002 | .181±.008 | .184±.008 | .184±.006 | .186±.005 |
| Bibtex | **.080±.005** | .082±.003 | .116±.006 | .326±.007 | .320±.008 | .106±.012 | .108±.004 | .120±.008 | .230±.004 |
| Medical | **.025±.008** | .031±.007 | .027±.008 | .085±.014 | .103±.016 | .050±.013 | .052±.019 | .043±.010 | .074±.011 |
| Birds | **.182±.037** | .192±.019 | .184±.035 | .312±.022 | .320±.018 | .427±.043 | .329±.045 | .352±.024 | .352±.024 |
| Emotions | **.189±.035** | .196±.022 | .233±.015 | .262±.057 | .263±.019 | .443±.022 | .345±.037 | .358±.021 | .358±.021 |
| *One Error* (the smaller, the better) | | | | | | | | | |
| Genbase | **.001±.002** | .001±.003 | .002±.003 | .013±.009 | .007±0.011 | .002±.003 | .004±.005 | .003±.004 | .015±.014 |
| CAL500 | .118±.018 | .123±.027 | .118±.020 | **.116±.018** | .282±.050 | .116±.019 | .126±.012 | .128±.033 | .117±.018 |
| Bibtex | **.366±.012** | .376±.013 | .373±.011 | .749±.023 | .588±.002 | .458±.017 | .389±.009 | .418±.013 | .634±.005 |
| Medical | **.142±.031** | .157±.011 | .164±.043 | .445±.065 | .234±.032 | .196±.040 | .241±.054 | .191±.050 | .283±.030 |
| Birds | **.424±.092** | .429±.056 | .445±.064 | .691±.050 | .619±.043 | .648±.041 | .806±.055 | .771±.078 | .745±.053 |
| Emotions | **.292±.059** | .320±.048 | .335±.052 | .418±.059 | .316±.040 | .391±.038 | .553±.051 | .479±.062 | .480±.038 |
| *Average Precision* (the larger, the better) | | | | | | | | | |
| Genbase | **.994±.006** | .993±.005 | .991±.004 | .966±.082 | .958±.036 | .980±.012 | .988±.005 | .980±.013 | .978±.013 |
| CAL500 | **.516±.013** | .513±.009 | .511±.013 | .511±.011 | .393±.016 | .500±.031 | .503±.016 | .496±.012 | .489±.010 |
| Bibtex | **.588±.006** | .579±.003 | .553±.007 | .474±.015 | .480±.005 | .486±.013 | .543±.013 | .488±.013 | .318±.006 |
| Medical | **.879±.021** | .877±.022 | .869±.032 | .646±.047 | .744±.026 | .833±.036 | .800±.048 | .839±.037 | .756±.024 |
| Birds | **.604±.012** | .592±.018 | .559±.048 | .403±.034 | .416±.019 | .441±.026 | .315±.040 | .360±.043 | .331±.026 |
| Emotions | **.776±.036** | .767±.024 | .742±.022 | .702±.022 | .737±.014 | .726±.015 | .579±.021 | .636±.031 | .631±.018 |

**Table 5: Comparision of PML-LENFN with other state-of-the-art PML and MLL algorithms on synthetic datasets and three evaluation metrics when** $r = 3$ **(mean±std), where the best results are in bold and the suboptimal are shown with an underscore.**

| Datasets | PML-LENFN | PML-DNDC | PML-SALC | PAR-MAP | PAR-VLS | fPML | PML-fp | LIFT | ML-KNN |
|---|---|---|---|---|---|---|---|---|---|
| **Ranking Loss** (the smaller, the better) | | | | | | | | | |
| Genbase | **.002±.003** | .003±.003 | .004±.003 | .017±.013 | .023±.006 | .009±.005 | .009±.003 | .011±.009 | .013±.010 |
| CAL500 | .177±.009 | .180±.009 | .179±.015 | **.176±.005** | .543±.018 | .182±.005 | .183±.009 | .185±.006 | .188±.008 |
| Bibtex | **.086±.006** | .088±.003 | .129±.006 | .335±.006 | .320±.007 | .108±.004 | .112±.006 | .124±.005 | .237±.003 |
| Medical | **.028±.008** | .031±.009 | .030±.008 | .089±.013 | .109±.018 | .049±.013 | .053±.013 | .043±.011 | .087±.015 |
| Birds | **.198±.045** | .201±.023 | .217±.028 | .337±.035 | .354±.025 | .391±.026 | .442±.051 | .358±.050 | .379±.022 |
| **One Error** (the smaller, the better) | | | | | | | | | |
| Genbase | **.001±.006** | .002±.004 | .004±.008 | .018±.013 | .008±.009 | .003±.003 | .019±.006 | .005±.006 | .016±.014 |
| CAL500 | **.116±.018** | .119±.058 | .117±.020 | .291±.047 | .117±.022 | .117±.022 | .127±.012 | .122±.018 | .117±.017 |
| Bibtex | **.371±.004** | .378±.010 | .386±.015 | .751±.018 | .588±.007 | .460±.007 | .396±.016 | .434±.011 | .640±.009 |
| Medical | **.149±.030** | .159±.001 | .164±.037 | .453±.074 | .280±.057 | .197±.042 | .255±.050 | .193±.042 | .312±.037 |
| Birds | **.438±.051** | .438±.054 | .468±.053 | .730±.055 | .617±.059 | .599±.044 | .817±.045 | .814±.065 | .778±.049 |
| **Average Precision** (the larger, the better) | | | | | | | | | |
| Genbase | **.992±.001** | .992±.005 | .989±.006 | .960±.011 | .957±.015 | .981±.013 | .982±.005 | .981±.013 | .974±.017 |
| CAL500 | **.515±.013** | .513±.018 | .510±.017 | .512±.011 | .397±.018 | .501±.011 | .503±.017 | .497±.013 | .481±.009 |
| Bibtex | **.579±.011** | .572±.007 | .533±.009 | .473±.006 | .479±.004 | .486±.008 | .530±.010 | .486±.009 | .312±.003 |
| Medical | **.876±.024** | .872±.021 | .865±.028 | .641±.053 | .708±.042 | .832±.033 | .789±.035 | .836±.032 | .728±.029 |
| Birds | **.593±.056** | .585±.036 | .561±.034 | .369±.039 | .337±.025 | .390±.029 | .289±.036 | .342±.042 | .338±.024 |

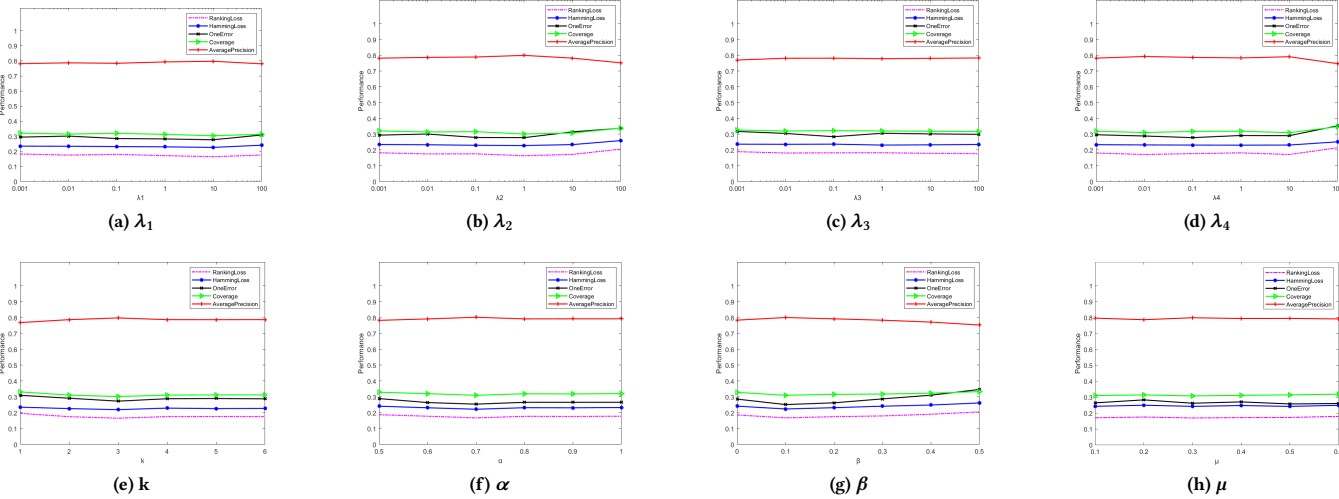

(a) *Ranking Loss*     (b) *One Error*     (c) *Average Precision*

**Figure 3: Results of PML-LENFN against other approaches with the Nemenyi test(CD = 2.9138 at 0.05 significance level).**

(a) $\lambda_1$   (b) $\lambda_2$   (c) $\lambda_3$   (d) $\lambda_4$

(e) k   (f) $\alpha$   (g) $\beta$   (h) $\mu$

**Figure 4: Results of Parameter sensitivity for PML-LENFN on Emotions.**

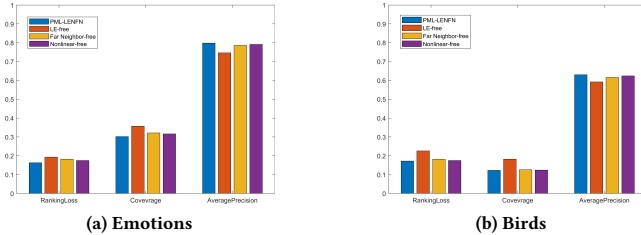

(a) Emotions    (b) Birds

**Figure 5: Ablation experiment on Emotions and Birds.**

## 5 CONCLUSION

In this paper, we introduce PML-LENFN, a novel partial multi-label algorithm. Firstly, it enhances labels using near and far neighbors information, richer than binary. The impact of label correlation on the model is also considered, with the correlation information from reconstructed labels transferred to the classifier. Additionally, nonlinear matrix decomposition is used to constrain the learning of the linear classifier in order to capture the model's nonlinear characteristics. Extensive experiments show the model's excellence. In future work, we will reduce the number of parameters and reference multi-view learning.

## 6 ACKNOWLEDGMENT

This work was supported in part by the National Natural Science Foundation of China under Grant No. 62176065, No. 62302172, in part by the Guangdong Provincial National Science Foundation under Grant No. 2022A1515011277.

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
