# OpenReview forum: "Partial Multi-label Learning Based On Near-Far Neighborhood Label Enhancement And Nonlinear Guidance"
_acmmm.org/ACMMM/2024/Conference — MM2024 Poster_

### Official Review · Reviewer_wEHE · 2024-05-14

[review text omitted: it was posted to a different submission]

---

### Official Review · Reviewer_LxVa · 2024-05-20

**Rating:** 5
**Confidence:** 4

**Summary:**

This paper introduces a novel method for partial multi-label learning (PML) named PML-LENFN. This method addresses the challenge of accurately predicting multiple labels for instances within noisy multi-label datasets. The core of PML-LENFN involves enhancing label information by incorporating both near and far neighbor information to mitigate noise, constructing a linear classifier to explore label correlations, and applying nonlinear mapping to capture the nonlinear relationships between features and labels.

**Strengths:**

- The PML-LENFN method is innovative as it combines label enhancement using both near and far neighbor information with nonlinear guidance, which is a unique approach in the field of PML.
- This paper presents a well-thought-out theoretical framework that includes a detailed explanation of how to reconstruct labels and how to integrate nonlinear mapping, which is crucial for capturing complex relationships in real-world data.
- The methodology described in the paper is technically sound, with a clear explanation of the optimization process using the ADMM method, which is a robust approach for solving such problems.
- This paper conducted a comprehensive evaluation using 17 synthetic PML datasets and 3 real PML datasets, demonstrating the effectiveness of PML-LENFN compared to existing methods.
- The paper is well-structured, and the authors have made efforts to clearly explain their approach, including the use of figures and tables to illustrate key concepts and results.

**Limitations:**

- The paper does not discuss the scalability of the proposed method to very large datasets. The computational complexity of the label enhancement process and the nonlinear mapping, especially when dealing with a high number of instances and features, could be a limitation.
- Experimental results show that PML-LENFN performs well on multiple datasets, but in the results analysis section, the paper does not delve into why the method works well in some cases and less well in others. It is recommended that the authors conduct a more in-depth analysis to understand the reasons for the changes in model performance.
- This paper mentions parametric sensitivity analysis, but does not provide detailed analysis results. It is recommended that authors provide a complete parameter sensitivity analysis so that readers can understand the effects of different parameter choices on model performance.
- The relevant work is mentioned in Section 2, but the discussion section does not seem to adequately compare and contrast PML-LENFN with existing methods. It is recommended that the authors discuss the advantages and limitations of PML-LENFN compared to existing methods in more detail in the discussion section.
- Some experimental results are provided in the paper, but the specific Settings of the experiment are not explained in detail, such as the pre-processing steps of the data set, the method of noise injection, and the number of repetitions of the experiment. In order to enhance the credibility of the experimental results, it is suggested that the authors provide more experimental details.

**Suitability:**

3

---

### Official Review · Reviewer_sQAF · 2024-05-21

**Rating:** 4
**Confidence:** 3

**Summary:**

The paper proposes a new method for partial multi-label learning (PML) called PML-LENFN. The method aims to address the challenges of utilizing label correlation and considering the nonlinear nature of the model in PML. It incorporates label enhancement based on near and far neighbors, a linear classifier based on label correlation, and matrix nonlinear decomposition.

**Strengths:**

1 The proposed method takes into account label correlation and the nonlinear nature of the model, which can lead to more accurate predictions.
2 The paper provides a detailed description of the proposed method, including the mathematical formulations and optimization process.

**Limitations:**

1 The motivation of this paper is not clear.
Although the authors have mentioned in the abstract that two problems existing in current works, their description is very vague.
2 The proposed method does not have a specific algorithmic flow description, which is difficult for a reader.
3 The organization of the paper's formulas is a bit confusing, with a mix of italics and bolding, and too many variables; it would be helpful to have a summary table with an enumeration of all the variables, which would enhance the readability of the paper.
4 Some formulations are confusing, such as  Eq. 3. The setting of these parameters is too arbitrary.

**Suitability:**

2

---

### Official Review · Reviewer_Dddr · 2024-05-24

**Rating:** 4
**Confidence:** 3

**Summary:**

This paper proposes a new Partial multi-label learning (PML) method based on label enhancement of near and far neighbor information and nonlinear guidance (PML-LENFN). It reconstructs the original binary label information by using the information of sample near neighbors and far neighbors to eliminate the influence of noise. It also constructs a linear multi-label classifier that can explore label correlation. Experiments demonstrate the superior performance of the proposed method. The main contributions of this paper are:

- It proposes a new label enhancement method, which reconstructs label information by using both near and far information of samples to achieve a denoising effect.
- It leverages the combination of linear and nonlinear training methods to train the classifier, which not only retains the linear characteristics of the model, but also adds the nonlinear characteristics to make it more consistent with the actual results.

**Strengths:**

- The problem studied in this paper is interesting.
- The paper is well-organized, which is easy to follow.
- The experimental results are somehow promising.

**Limitations:**

- In Section 1, although the authors mention, "in practice, the models we need to fit are basically curvilinear structure", this statement needs to be verified through experimental observation or theoretical analysis.
- In Section 4.4, the authors mention, "we found that the model is not sensitive to $\lambda_1$ , $\lambda_3$, and $k$ . In this model, the weight of label correlation constraints $\lambda_2$ , the weight of nonlinear constraints $\lambda_4$ and the weight of distant information $\beta$ cannot be too large". The reasons for these phenomena should be further analyzed.

**Suitability:**

2

---

### Meta-Review · Area_Chair_KT6A · 2024-07-06

**Recommendation:** Accept (Poster)
**Confidence:** 5

**Metareview:**

Initially, all four reviewers appreciated the paper’s contribution. They appreciated the interesting idea, and the demonstrated improvement over the prior art. The rebuttal addressed the reviewers’ concerns. All four reviewers recommended that the paper is above the bar for acceptance to MM. The AC agrees with their recommendation. Please take into account all reviewer feedback in the camera-ready version.